# Screening Risk Assessment of Agricultural Areas under a High Level of Anthropopressure Based on Chemical Indexes and VIS-NIR Spectroscopy

**DOI:** 10.3390/molecules25143151

**Published:** 2020-07-09

**Authors:** Agnieszka Klimkowicz-Pawlas, Guillaume Debaene

**Affiliations:** Department of Soil Science Erosion and Land Protection, Institute of Soil Science and Plant Cultivation—State Research Institute, Czartoryskich 8, 24-100 Puławy, Poland; gdebaene@iung.pulawy.pl

**Keywords:** visible and near-infrared spectroscopy, ecological risk assessment, soil contamination, agricultural soils, PAHs, anthropopressure, risk indexes

## Abstract

Intensive anthropogenic activity may result in uncontrolled release of various pollutants that ultimately accumulate in soils and may adversely affect ecosystems and human health. Hazard screening, prioritisation and subsequent risk assessment are usually performed on a chemical-by-chemical basis and need expensive and time-consuming methods. Therefore, there is a need to look for fast and reliable methods of risk assessment and contamination prediction in soils. One promising technique in this regard is visible and near infrared (VIS-NIR) spectroscopy. The aim of the study was to evaluate potential environmental risk in soils subjected to high level of anthropopressure using VIS-NIR spectroscopy and to calculate several risk indexes for both individual polycyclic aromatic hydrocarbons (PAHs) and their mixture. Results showed that regarding 16PAH concentration, 78% of soil samples were contaminated. Risk assessment using the most conservative approach based on hazard quotients (HQ) for 10 individual PAHs allowed to conclude that 62% of the study area needs further action. Application of concentration addition or response addition models for 16PAHs mixture gave a more realistic assessment and indicates unacceptable risk in 23% and 55% of soils according to toxic units (TUm) and toxic pressure (TPm) approach. Toxic equivalency quotients (TEQ) were below the safe limit for human health protection in 88% of samples from study region. We present here the first attempt at predicting risk indexes using VIS-NIR spectroscopy. The best results were obtained with binary models. The accuracy of binary model can be ordered as follows: TPm (71.6%) < HI (85.1%) < TUm (87.9%) and TEQ (94.6%). Both chemical indexes and VIS-NIR can be successfully applied for first-tier risk *assessment*.

## 1. Introduction

Anthropogenic activities such as industrialization, urbanization, mining and agriculture intensification can generate serious soil contamination problems worldwide. Soil pollution has been identified as the third most important threat to soil functions in Europe [1] and raises serious concerns about the risks to human and ecosystem health. The presence of both inorganic and organic pollutants may strongly affect different soil functions. One group of contaminants of special concern are polycyclic aromatic hydrocarbons (PAHs), semi-volatile, stable, hydrophobic compounds that are ubiquitous in the environment and are good markers of anthropogenic pressure [2,3,4]. Anthropogenic activities including biomass combustion, domestic and industrial coal combustion, coking industry, and emissions from road vehicles are major sources of PAH contamination; all these sources represent continuous inputs of PAH contaminants [5,6,7,8]. After deposition on soil surfaces, PAHs may further accumulate in plants (e.g., vegetables) and other biota and be transferred to humans via the food chain, or they can strongly sorb on soil, where they can persist for long periods of time [9,10].

PAH-contaminated agricultural soils have attracted considerable public attention since they can contaminate food, which is a threat to human health. Therefore, the identification and remediation of contaminated sites is of increasing importance [3,11]. The need for remediation is evaluated through decision-support systems based on assessment of the risk to human health, the soil ecosystem and food safety. Generally, ecological risk (ERA) evaluation is generic or site-specific, depends on future land management, and must include the identification of contaminants and a risk assessment of potential exposure [12,13]. For optimum protection of the soil ecosystem, risk assessments should start with the application of generic and conservative assumptions. Screening first tier ERA often includes the use of various quotients or indexes, where actual or predicted concentrations of chemicals in soils are compared with national environmental quality standards (e.g., soil screening levels or benchmark values) or toxicity measures (e.g., no-observed-adverse-effect concentrations). In most cases, hazard screening, prioritisation and subsequent risk assessment are performed on a chemical-by-chemical basis. As a first screening, the following indexes are frequently used: hazard quotient (HQ), toxic units (TU), toxic pressure (TP) or toxicity equivalency factor (TEF) methodology [3,14,15,16,17].

Traditional methods of hazard assessment of PAH-contaminated sites involve soil sampling, extraction of PAH compounds using various extraction techniques and solvents, and analysis of the liquid extracts by gas chromatography-mass spectrometry (GS/MS) [16]. Conventional methods of contaminant identification (although very accurate) are costly and time-consuming [12]. Therefore, there is a need to look for fast and reliable methods for risk and contamination prediction in soils. Among such innovative methods is visible and near-infrared (VIS-NIR) reflectance spectroscopy—a rapid, non-destructive, reproducible, environmentally friendly technique involving diffuse reflectance measurements in the visible (VIS: 350–780 nm) and near-infrared (NIR: 781–2500 nm) regions [18,19]. Reflectance signals result from vibrations in compounds’ covalent chemical bonds (e.g., C-H, O-H, N-H), and provide qualitative and quantitative information about the proportion of each element in the analysed sample [18]. The energy absorption by hydrocarbon derivatives is due to overtones and combinations of fundamental vibrational C-H stretching modes of saturated CH_2_ and terminal CH_3_, or aromatic C-H groups [20]. Therefore, VIS-NIR gives information on the structure of organic matter since it is based on the absorbance of different bonds found in proteins, cellulose, carboxyl, amide or amino acids [21,22]. 

VIS-NIR spectroscopy is now an established method for the prediction of several basic soil properties [20]. It was successfully used for the quantitative analysis of a variety of physical and chemical soil properties such as moisture, total C, N and P content, quality of organic matter, etc. [20,23], and soil biological parameters such as respiration, microbial biomass, potential mineralization of N and ratio of microbial C to organic C [18]. VIS-NIR spectroscopy is gradually becoming more popular for the rapid screening of contaminants in soils as more recent studies have demonstrated its efficacy in the detection of heavy metals (Cd, Pb, Cu, Zn) [24,25], total and bioavailable PAHs [12,26,27,28], individual compounds, e.g., benzo[a]pyrene [12] or phenanthrene [29], and total petroleum hydrocarbons [30,31]. The method provides a useful alternative to classical time-consuming chemical methods of soil contamination analysis [24] and can be used in situ [32]. However, to date, only a small amount of literature can be found regarding the use of VIS-NIR spectroscopy for screening risk assessment and to predict PAH toxicity in soil environments. Okparanma et al. [16,26] used this technique for prediction of PAH concentrations in petroleum- contaminated tropical rainforest soils in Nigeria, and for mapping of PAH and their potential toxicity (as total toxicity equivalent concentration).

The main aim of our study was to evaluate the potential ecological risk for agricultural areas under a high level of anthropogenic pressure. Screening risk assessment included: (1) evaluation of the contamination status of the study area; (2) calculation of several chemical indexes using different approaches to characterise both ecological and human health risk; and (3) application of VIS-NIR spectroscopy for analysis of PAH concentrations and environmental risk prediction.

## 2. Results and Discussion

### 2.1. Characteristics of Soil Properties and PAHs Concentrations

The collected soils were dominated by loamy sands with 70.7%, 26.8% and 1.9% of sand, silt and clay, respectively (Table 1). Most of soils exhibited relatively high acidity (median for pH_KCl_ was 5.2; interquartile range 4.6–5.6).

The content of TN and C_org_ was in the range of 0.2–11.7 g kg^−1^ (CoV 112%) and 5.2–187.2 g kg^−1^ (CoV 142%), respectively. However, in 75% of soils, TN did not exceed 1.4 g kg^−1^, and C_org_ was below 17.6 g kg^−1^, which corresponds to the lower range according to the European Union criteria [33]. The low content of organic matter and clay promotes the increase in contaminant availability, which may result in disturbance of soil functions and create risk to soil organisms [11].

The total concentration of 16PAHs varied strongly throughout the sampling area (CoV = 410%), and ranged from 311 to 316.1 × 10^3^ µg kg^−1^, with a mean value of 11.8 × 10^3^ µg kg^−1^ and a median of 1252 µg kg^−1^ (Table 1). Average PAHs content significantly exceeded the values reported for the agricultural soils from non-industrial areas in Poland (291–616 µg kg^−1^ [4,5,34]), Spain (112 µg kg^−1^ [35]), Norway (63 µg kg^−1^ [36]), Italy (333 µg kg^−1^ [37]), as well as in China (202–219 µg kg^−1^ [6,38]). However, our results were comparable to the PAHs concentration reported for agricultural soils in Czech Republic (847 µg kg^−1^ [39]), for soils from rural regions in the surroundings of coking plant (691 µg kg^−1^ [40], 3016 µg kg^−1^ [7]) and mining areas (1280 µg kg^−1^ [41], 917 µg kg^−1^ [42]) in China. Similar high levels of PAHs were also observed in forest soils from the Silesia region in Poland (1050 µg kg^−1^ [8]), as well as in urban soils from Spain (1002 µg kg^−1^ [35]), Portugal (1544 µg kg^−1^ [2]), Slovakia (2065 µg kg^−1^ [43]) and Poland (1331 µg kg^−1^ [44]).

Contamination status of the area was characterized using previously developed classification system proposed by Maliszewska-Kordybach [45] and Terelak et al. [46]. The 16PAH concentrations were classified as follows: background value (<200 µg kg^−1^, class 0), non-contaminated (200–600 µg kg^−1^, class 1), moderately contaminated (600–1000 µg kg^−1^, class 2), contaminated (1000–5000 µg kg^−1^, class 3), heavily contaminated (5000–10,000 µg kg^−1^, class 4), and very heavily contaminated (>10,000 µg kg^−1^, class 5). According to these criteria, 78% of soil samples should be classified as contaminated to varying degree, with PAH concentrations ranging from 1000 to 5000 µg kg^−1^ in 50% of samples, and PAH > 5000 µg kg^−1^ in 12% of samples. The highest PAH levels were found in the samples from central and north parts of the Czerwionka region and might be attributed to PAH emission sources: historical long-term coke production and coal mining, and nowadays production of bituminous products, waste recovery, high traffic volume and house heating [11].

The high molecular weight hydrocarbons (HMW; ≥4 rings) dominated the PAHs profile in the region (Table 1) and ranged from 180 to 270 × 10^3^ µg kg^−1^. 54% of the HMW pool were compounds considered to be carcinogenic (ΣPAH_Carcin_), and 30% four hydrocarbons (BbF, BkF, BaPyr, IndPyr), that are monitored within the European Monitoring and Evaluation Programme of Air Emissions [47]. The annual emission of ΣPAH_4em_ in the study area is estimated at 1.75 kg per km^2^, and the dust emission (another anthropogenic index) at 6827 kg per km^2^ [4]. The predominance of HMW is typical in areas of high anthropogenic pressure [2,4,6,7,42] and may be due to the higher persistence of these compounds in soils, a dominance of combustion over petrogenic sources, and the tendency of HMW PAHs to accumulate in soils that are close to emission sources [2,4,5,10,36,44].

The PAHs isomer ratio of individual compounds is a common tool to identify sources of these hydrocarbons [4,7,10,37]. Three of the most stable molecular ratios were used in this study (Table 1). The Fln/(Fln + Pyr) ratio ranged between 0.41 to 0.67 and pointed to coal and biomass combustion in almost 99% of samples. Similarly, the BaA/(BaA + Ch) ratios above 0.35 indicated combustion of vegetation and fossil fuel as a PAH source. This was also confirmed by the IndPyr/(IndPyr + BPer) ratio > 0.5, however, in 36% of samples this ratio ranged from 0.2 to 0.5, which is more typical for liquid fossil fuel combustion (vehicle and crude oil) [4,37].

### 2.2. Evaluation of Potential Ecological Risk—Chemical Indexes

Potential ecological risk was firstly assessed using hazard quotient values calculated for 10 individual PAHs included in Polish soil regulation [48]. According to Polish guidelines, the acceptable maximum permissible concentration (MPC) for Anth, Ch, IndPyr and BPer is 200 μg kg^−1^, while for Napht, BaA, BbF, BkF, BaPyr and DahA, the MPC is 100 μg kg^−1^. HQ values of individual PAH compound > 1 indicate a possible ecological risk [11,17,49]. Average HQ value of a single chemical (expressed as median) varied significantly depending on PAH compound from 0.06 for Anth to 1.32 for BbF (Table 2), what allows us to conclude that according to individual PAH, 7 to 62% of the study area might be under ecological risk (HQ above 1), and needs further actions (e.g., site-specific evaluation or remediation).

To reflect the potential risk caused by PAHs mixture the hazard index (HI) was additionally calculated, and to assess the degree of potential risk the approach of Moreno-Jiménez et al. [17] was used. As a threshold value for HI index above which the possible risk occurs—the value of 10 was assigned [11,14]. Obtained HI values ranged between 1.56 and 1523 (Table 2) but exceeded the threshold for HI only in 23% of soil samples from the study region. The moderate risk (HI = 10–100) and high risk (HI > 100) were noted in 13 and 4 soil samples, respectively.

Many authors have emphasized that the HQ-based methodology, although frequently used for screening risk assessment [17,49], as a very conservative approach, which may lead to risk overestimation [3,9]. Furthermore, in our study HQ/HI indexes were calculated only for 10PAH compounds to thus better predict ecological risk in the study region, other approaches were applied and toxic units and toxic pressure were also calculated.

The model of toxic units is frequently used in ecotoxicology and represents the ratio between the concentration of a component in the mixture and its toxicological endpoint, e.g., effect concentration (EC_50_) or non-observed effect concentration (NOEC) [3,49,50,51]. Toxic units of the mixture (TUm) is calculated based on the concentration addition model (CA), which is based on the assumption that the mixture components possess a similar toxicological mode or mechanism of action, and in the case of PAHs it is caused by narcosis [50,51]. In our study, the measured soil PAH concentrations were compared to PNEC values for 16PAH compounds (Table 3). PNECs were adapted from the EU risk assessment report for coal-tar pitch [3,52], which were derived by applying an assessment factor to the lowest available NOEC values for the most sensitive terrestrial species, e.g., *Folsomia candida*, *Folsomia fimetaria*, *Lactuca sativa* or nitrifying bacteria (depending on the PAH compound). Although such deterministic approach based on only one species is rather conservative, it is a good protective measure and it is suggested for risk assessment of chemical mixtures in a first-tier of ERA process [51,52]. The TU values calculated for each individual PAH and TUm for 16PAHs can be found in Table 3.

Toxic units higher than 1 indicate that unacceptable effects on soil organisms are likely to occur. The TU for individual hydrocarbons varied significantly (CoV from 325 to 465%) and on average was 0.12 for Flu up to 20.5 for BaPyr. Generally, TU above 1 was noted mainly for higher molecular PAH compounds, an unacceptable effect on soil organisms may occur in 22–23% of soil samples contaminated with IndPyr, BPer and BbF, and in 43 and 66% of soil samples due to contamination by BaA and BaPyr, respectively. For other PAHs, the risk predicted with TU was acceptable (TU < 1) in 88–96% of soil samples (Table 3). The toxic units calculated for the 16PAHs mixture (TUm) ranged from 1.2 to 1858 with a median value of 5.7, which indicated that the exposure to the 16PAHs mixture may reveal the risk to soil organisms in 100% of soil samples (Table 3). For the rating of the TUm values and to differentiate the magnitude of risk to soil organisms, the classification system elaborated by Persoone et al. [53] was adapted, according to which soil samples with TUm < 1 reveal no toxicity, when TUm is between 1-10, samples are considered as lowly toxic, while samples with TUm above 10 and 100 could be classified as highly toxic and very highly toxic, respectively. It was found that TUm values for the majority (77%) of soil samples ranged between 1.26 to 9.56 and could be considered as lowly toxic; the highest TUms above 100 were only in four sampling points (5.4% of the study area) located in the surroundings of the main contamination sources—coking plant and coal mining landfill [11].

The third approach used to assess potential ecological risk to the environment was based on toxic pressure (TP) calculation for both individual compounds, as well as for the 16PAHs mixture (Figure 1, Table 4).

The response addition model suggested by Jensen et al. [13] was applied for TP of mixture determination. This model assumes dissimilar modes of action for all compounds [13], although, the same mode of action is normally considered for chemicals like PAHs [3]. TP for single chemicals were calculated using the soil screening values as the effect level. Since according to Polish soil guidelines, screening values are available only for 10PAHs, to calculate toxic pressure for the 16PAHs, the maximum permissible concentrations (MPC, Table 4) from the Dutch regulations were adopted [3,54]. MPC values reflect PAH concentrations that should protect all species in the ecosystem from adverse effects. In our study, TPs obtained for individual hydrocarbons were rather low, and ranged on average from 0.03 for Flu to 0.26 for BaPyr (Figure 1). To assess the magnitude of risk that pollutants may pose to the soil organisms, TP classification according to Jensen et al. [13] and Dagnino et al. [15] were applied. Considering the TP of single chemicals, the majority of samples (66–96%) have TP values below the lowest threshold 0.25 (Table 4), which indicates no risk in this area. The highest TPs were noted for higher molecular weight (>4 ring) PAH, and can be ordered as follows: BaPyr > BaA > BbF > IndPyr > BPer > Pyr > DahA = Ch = BkF. Completely different results were obtained when TPs were calculated for the 16PAH mixture, TPm ranged from 0.0 to 1.0 with an average of 0.53 (Figure 1). The lowest TPm < 0.25 (no risk) was found in only 34% of soil samples (Table 4, Figure 2), in 11% of samples TPm ranged 0.34–0.50 (low risk), in 20% of samples between 0.51 and 0.74 (moderate risk), and as much as 35% of soil samples had TPm value above 0.75 (high risk). Extremely high TPm values (above 0.9) were observed in 16 soil samples from the study region. However, it should be remembered that TP index in our study reflects the total PAHs concentration and is based on the conservative assumption that 100% of measured contaminant concentration is bioavailable [9,11,13]. On the other hand, it is considered that such conservative approach (highly protective for soil organisms) should be used in a generic first tier risk assessment [3,51].

### 2.3. Evaluation of Potential Human Health Risk—Chemical Indexes

Toxic equivalency factors provided by Nisbet and LaGoy [55] were used in our study to estimate the exposure risks posed by individual compounds and mixture of PAHs to human health. The toxic potency of each PAH (TEQ) was assessed on the basis of its benzo(a)pyrene equivalent concentration [3,7,37,55,56] and for 16PAHs ranged from 21 to 52 × 10^3^ µg kg^−1^ with a median value of 151 µg kg^−1^ (Table 5).

Our results were comparable to toxic PAH concentrations reported for industrial and residential regions in Poland [8,44], Spain [35] and Italy [37], as well as for rural regions under high anthropogenic pressure in China [7,42]. Results highlighted a significant variation of TEQs values (CoV up to 465%) through the study region (Figure 3). For the majority of samples TEQs ranged from 21 to 584 µg kg^−1^ and were below the safe value of 600 µg kg^−1^ established as a limit for protection of human health (regardless of the land use) in Canadian soil quality guidelines [3,57], which indicated a generally low risk in about 88% of samples from Czerwionka region. To better evaluate our results, we compared TEQ values with other guidelines values available for BaPyr [3,57,58]. All TEQ in our study exceeded the most conservative US EPA soil screening level of BaPyr for residential areas (15 µg kg^−1^ [3]). About 38% had the TEQ below 100 µg kg^−1^, and 31% of samples < 200 µg kg^−1^, which corresponded to values for residential areas in Italy and Spain [3,58], respectively. The highest toxicity value of PAHs (TEQ above Danish cut-off criteria 1000 µg kg^−1^ [3]) in soils was concentrated in the central and north part of the Czerwionka region (Figure 3) in the immediate vicinity of the main contamination sources (coke plant, asphalt production plant, post-mining landfill and coal recovery plant).

It is worth emphasizing that only in one sampling point TEQ exceeded 2000 µg kg^−1^ corresponding to the Finnish guideline value [58] which considers a target total lifetime cancer risk (TLCR) of 10^−5^, and only in three samples TEQ was above 10,000 µg kg^−1^ (Italian screening value for commercial and residential areas, [3]).

### 2.4. Application of VIS-NIR for Potential Ecological and Human Risk Prediction

Until now, VIS-NIR spectroscopy was used for total and bioavailable PAHs, alkanes and heavy metals determination [27], alkanes and PAHs in oil-contaminated soil [28] or the bioremediation of soils polluted with fuel oil [31]. In our study, this technique was used for diagnostic risk screening of long-term PAH contaminated soils, and to predict their toxicity based on various risk indexes.

The obtained spectra (Figure 4) are typical for mineral soils with a high sand content (loamy sand) [59] but are relatively flat with no clear features. This is the result of sample drying and sieving but also of the relatively low C_org_ content [20,32] of most samples (upper quartile for C_org_ 17.6 g kg^−1^). The differences between samples are mostly noticeable in baseline and peak intensities. The reflectance is usually increased with a lower SOM content [60]. In the present study, C_org_ is probably not the only factor governing the reflectance intensities since the mean C_org_ content from class 1 (200–600 μg kg^−1^) to class 5 (>10,000 μg kg^−1^) are 9.56; 9.81; 22.96; 17.33, and 87.94 g kg^−1^, respectively. The absorption features at 1400 and 1900 nm are due to water (OH bonds) and the absorption feature at 2210 nm is due to hygroscopic water in clay minerals [61]. Overall, there was a reflectance decrease with the increase of PAHs content. This was also observed by Bray et al. [12] and Okparanma et al. [16,26], and could partly be explained by the correlation between soil organic carbon and 16PAHs.

To investigate further the structure hidden within the soil spectra, a principal component analysis (PCA) was undertaken. The score plot is presented in Figure 5. Most of the spectral variations (99%) are described by the first component (PC-1). Samples are distributed along the PC-1 axis. Two cluster are well separated (low PAH content (class 1) and high PAH content (class 5). Variations of C_org_ content and soil texture in the other classes were too great to allow for clustering according to PAH content. Bray et al. [12] on suburban soil samples have noticed that samples were roughly distributed along PC-1 axis. That behaviour could be related to soil properties (e.g., C_org_ content).

One sample (down the PC-2 axis) seems to be an outlier, but was not removed from the models since preliminary investigations have shown that it did not have any significant effect on the modelling. That sample was an organic soil (muck) while the other samples were mineral soils. The green triangles (Figure 5) are soils from the 16PAH 1000–5000 μg kg^−1^ class. Their spread is probably due to the fact that they present most of the soil texture variation and texture affects soil reflectance [62], but also the most C_org_ variation.

The texture heterogeneity leads to more spectral variation related to diffuse reflected light inside the class than the other PAH classes. No features in the 1600–1800 nm related to hydrocarbon contamination [16,26] were noticeable in the raw spectra probably due to the fact that samples were sieved and scanned dry. However, even if most features are not discernible to the naked eye, chemometric techniques (e.g., PLS) are able to extract important information from a spectral database.

The PLS regression models of individual PAHs were poorly defined with r^2^ < 0.5 and high root mean square errors and are not reported here. Models with MA pre-processing were better than with other pre-processing methods and are the only ones presented below. The quantitative predictions of individual PAHs and 16PAHs were not satisfactory probably due to the low representation of extreme values and to the limited number of samples. The non-normal distribution of PAH contents is also a factor affecting the predictions. To our knowledge, only one paper predicted an individual PAH (BaPyr) concentration with success [12] on 65 urban soils from Australia. However, their BaPyr content was much higher than in our dataset (74 samples). Phenanthrene, as an individual PAH was also predicted in another study but on artificially contaminated soil samples with a very large amount of diesel oil [29]. The lack of robust prediction for individual PAH is probably due to an insufficient number of samples representing each class of contamination and to a relatively low PAHs content (except for a few samples). Despite good correlation with C_org_ (r^2^ = 0.81 for the five composite samples from Figure 5), PAH contents were not possible to predict. Nevertheless, the grouping of samples along the PC-1 component leads us to believe in the possibility to classify samples according to the different risk indexes presented in the above sections.

Figure 6 presents a plot of PLS regression coefficients obtained by the PLS regression of total 16PAH. That plot allows identifying important variables (wavelengths) of the model construction.

Assignment to specific compounds here is difficult too. No distinctive peaks were identified as important but rather regions (thick black line). Those regions are mostly related to organic matter (OM) and clay minerals. The first region (560–804 nm) is related to chromophores of OM and iron oxides [20,63]. The second region (1517–1898 nm), which is the first overtone region is mostly related to OM and contains the absorption bands by several bonds (CH, CH_2_, CH_3_, and Ar-CH aromatic ring [64]). Moreover, Okparanma et al. [16] identified a part of that region (1600–1800 nm) as related to hydrocarbon contamination. That important variables region is discontinued before the 1900 nm absorption peak related to water. The next region (1983–2046 nm) starts just after that peak and is related to N-H bonds and aromatic amines [64]. The last region (2188–2236 nm) is probably related to the 2210 nm absorption band from hygroscopic water and clay mineral [61].

We, therefore, conclude that predictions of risk indexes (especially binary models) by VIS-NIR spectroscopy are possible because of the PAH relationship with OM but also with clay and perhaps iron oxides (see Figure 6). Since individual PAHs were poorly predicted, we decided to run the PLS algorithm on the index classes to use the method as a fast screening tool to predict environmental risk. Classes (e.g., 1 to 4 for TUm) were predicted instead of actual index values. Table 6 summarises the prediction of the classification results.

Prediction of Σ16PAHs classes was accurate in only 58.1% of the samples (Table 6). The model could not correctly classify samples from the 200–600 μg kg^−1^ class (non-contaminated samples). We, therefore, divided the dataset into two classes (threshold—600 μg kg^−1^) to test if it was possible to predict if a sample was contaminated or not. In that case, prediction accuracy was 79.7% (59 correctly classified samples out of 74). Still, the model had difficulties in detecting samples of the non-contaminated class (11 cases were predicted as contaminated but being non-contaminated).

Besides, the prediction of the soil PAH concentrations, the VIS-NIR method was also used to predict environmental risk. The prediction of hazard index (HI) classes was better with 83.8% accuracy. The model very accurately predicted samples of low impact (1 < HI < 10), where HI index did not exceed threshold value for acceptable risk. Out of the 12 samples wrongly classified, four were classified as class 2 (HI = 10–100, moderate risk) instead of class 1 (HI = 1–10, low risk), six samples were classified as class 1 instead of class 2, and two samples classified as class 2 instead of class 3 (HI > 100, high risk). When considering for a binary prediction only HI index above which the possible risk occurs (HI > 10), the accuracy was 86.5% with eight samples classified as class 2 (HI > 10) instead of class 1 (HI < 10) and only two samples the other way around.

The classification of toxic units for 16PAH mixture (TUm) according to Persoone et al. [53] has provided satisfactory results with an accuracy of 83.8%. When predicting if a sample presented toxicity (TUm > 10) with a binary prediction, the accuracy was 87.9% with no over prediction of one class over the second. This is in agreement with Bray et al.’s [12] accuracy of 90.2% for the prediction of Σ16PAHs > 10 mg kg^−1^ using ordinal logistic regression.

The TP class predictions were unsatisfactory, with only 33.8% accuracy classifying the samples into the four TP classes [15]. The best predicted class was the low risk class (0.25 < TP < 0.50) with 75% accuracy. Nevertheless, there was rarely more than one class error difference (e.g., class “no risk” instead of class “moderate risk”). A binary prediction with a threshold of 0.5 was completed with an aim at detecting if there is a risk or not [15], and the obtained prediction risk accuracy was 71.6%. It needs to be highlighted that more investigations are needed to develop more accurate models of TP classification when using the 4 classes by including more samples of each class.

The prediction of TEQ classes were made separately for each threshold according to Carlon et al. [58] and Cachada et al. [3]. There were no samples of TEQ < 15 μg kg^−1^, therefore seven independent models were produced (Table 6). There was an increase in prediction accuracy with increase of the TEQ threshold value. Bray et al. [12] obtained an opposite pattern, with lower accuracy while predicting higher threshold values (but for Σ16PAHs; 10, 20, and 40 mg kg^−1^). In our case, this is related to the fact that there were fewer samples with higher contamination values, and thus higher TEQs. For a TEQ threshold of 600 μg kg^−1^, which is the limit Canadian value for the protection of human health [57], the binary model (if a sample has a TEQ higher or lower than 600 μg kg^−1^) accuracy is 94.6%.

There was overestimation in the predictions of 16PAHs and HI index but overall, our results are promising and the method could be of interest to use a fast screening tool to decide whether or not more investigations are needed in a specific area. This is the first time that VIS-NIR was used to predict HI, TUm, TP, and TEQ indexes in a view to allow a fast screening risk assessment of a contaminated area.

## 3. Materials and Methods

### 3.1. Study Area Characterization

The area considered in this study covers a territory of 115 km^2^ located in the south-western part of the Silesian Province in Poland (Rybnik district, Czerwionka municipality). This region is rich in natural resources, including hard coal, rock salt, and construction sand. The research area combines typical urban, industrial and post-industrial character districts, as well as forests and agricultural areas of high natural and recreational value. The land usage structure is dominated by forest areas (42%) and arable lands (34%). The soil cover is represented by Cambisols and Luvisols with small contributions of Phaeozems, Fluvisols and Histosols. Soil contamination in this region is mainly the result of fossil fuel combustion, waste recovery, road transport, previous coal mining activity and long-term (for over 100 years) coke production. A detailed description of the research area can be found in Klimkowicz-Pawlas et al. [11].

The soil samples (n = 74) were collected from the surface layer (0–30 cm) of agricultural land (mainly arable fields—75%) in two sampling campaigns after the vegetation season and analysed for physical, chemical and spectral properties, as well as for the content of selected contaminants (PAHs). Geographical locations of sampling points were preliminary based on maps (1:25,000 scale; database of the Institute of Soil Science and Plant Cultivation in Puławy) and identified by a GPS technique. The sampling sites reflected direct exposure to local and transboundary pollution emission sources and variability of soil and hydrological conditions. Usually, the sampling density was higher in area of intensive anthropogenic activity and lower in recreational/forested areas. The soil material was air-dried at 20 °C, well mixed, sieved to pass a 2 mm sieve-mesh and stored in the dark at a temperature of 16–18 °C before chemical and spectral analyses.

### 3.2. Soil Analysis

Physicochemical characteristics of soils included particle size distribution, pH, total organic carbon and total nitrogen content, as well as PAH concentrations. Particle size distribution was measured by the laser diffraction method, using a Mastersizer 2000 apparatus with a Hydro UM attachment (Malvern Panalytical, Malvern, UK,) [65]. Soil reaction was determined potentiometrically in a suspension of 1 mol L^−1^ KCl (1:2.5 (m V^−1^)). Total organic carbon (C_org_) content was measured by sulfochromic oxidation followed by titration of the excess K_2_Cr_2_O_7_ with FeSO_4_(NH_4_)_2_SO_4_·6H_2_O, while total nitrogen (TN) by the Kjeldahl method.

Determination of PAHs included 16 priority compounds from the US EPA list: naphthalene (Napht), acenaphthylene (Acyn), acenaphthene (Acen), fluorene (Flu), phenanthrene (Phen), anthracene (Anth), fluoranthene (Fln), pyrene (Pyr), benzo[a]anthracene (BaA), chrysene (Ch), benzo[b]fluoranthene (BbF), benzo[k]fluoranthene (BkF), benzo[a]pyrene (BaPyr), indeno[1,2,3-cd]pyrene (IndPyr), dibenz[a,h]anthracene (DahA), and benzo[ghi]perylene (BPer). Soil PAHs analysis followed the procedure described elsewhere [4,10,34]. Soil samples (grain size ≤ 0.1 mm) were spiked with a mixture of five deuterated PAH compounds, extracted with dichloromethane in an Accelerated Solvent Extractor (ASE 200, Dionex Co., Sunnyvale, CA, USA) and analysed by gas chromatography with MS detection using Agilent GC-MS system (Agilent Technologies, Santa Clara, CA, USA). The quality control procedure included the analysis of certified reference material (CRM 131), laboratory control samples and solvent blanks. The recovery for individual compounds from CRM 131 was within 62–84% and the precision expressed as a relative standard deviation (RSD) was in the range of 5–12%. Limit of detection (LoD) for individual PAH compounds was within 0.5–2.3 μg kg^−1^, while limit of quantification (LoQ) was 1.6–7.1 μg kg^−1^.

### 3.3. Potential Ecological and Human Health Risk Calculation

The risk assessment was based on the total concentration of contaminants and the calculation of risk indexes for individual chemicals and for their mixture; the hazard index (HI), toxic pressure (TP), toxicity equivalency factor quotient (TEQ), and toxic units (TUm) were used. HI was expressed as the sum of hazard quotient values (HQs) calculated for individual PAH compounds and reflecting a ratio between the total measured concentration of PAH in soils (C_PAH_ in µg kg^−1^) and the maximum permissible concentration (MPC in µg kg^−1^) for individual compound. The hazard index was calculated as follows [3,11]:(1)$$HI=\Sigma HQ=\Sigma \frac{CPAH}{MPC}$$

The toxic unit (TU) model described previously by Gómez-Gutiérez et al. [49] and Cachada et al. [3] was used to calculate ecological risk. The soil PAH concentration (C_PAH_ in µg kg^−1^) was compared with the predicted no-effect concentration (PNEC in µg kg^−1^). PNEC is defined in the literature as “the concentration below which unacceptable effects on organisms will most likely not occur” [3]. Since for pollutants like PAHs it is assumed that individual compounds have the same toxic mode of action, the individual TUs of the 16PAHs were summed obtaining the toxicity of the mixture (TU_m_) and calculated as follows:(2)$${\mathrm{TU}}_{m}=\sum \mathrm{TU}=\sum \frac{C\;PAH}{PNEC}$$

In addition, the Toxic Pressure coefficient (TP_m_) for the mixture was calculated using the approach suggested by Jensen et al. [13] and Dagnino et al. [14], based on the response addition model, which assumed a dissimilar toxic mode of action for all compounds. The following formula was used for TP_m_ calculation:

TPm = 1 − (1 − TP_1_) (1 − TP_2_) (1 − TP_3_) ……… (1 − TP_n_)
(3)
where TP_1_, TP_2_, TP_3_, TP_n_ is the toxic pressure calculated for the individual PAH compounds [3].

The toxic equivalent factors (TEFs) are an example of relative potency factors (RPF) and were used for the toxic equivalency quotient (TEQ in µg kg^−1^) derivation according to the following formula [56]:

TEQ = ∑(C_PAH_ × TEF)
(4)
where TEQ is obtained by summation of the products of the concentration of individual PAH (C_PAH_ in µg kg^−1^) and corresponding TEF value. A TEF is estimated based on the relative toxicity of a chemical compared to a reference chemical, which for PAHs is benzo[a]pyrene (BaP). The TEF value of 1 is assigned to the BaP, for the other chemicals TEFs are order-of-magnitude estimates of potency.

### 3.4. Environmental Risk Prediction Using VIS-NIR Spectroscopy

Grounded and sieved samples (2 mm) were analysed with a PSR-3500 spectroradiometer (Spectral Evolution, Haverhill, MA, USA). The instrument acquires reflectance data in the 350–2500 nm range with spectral resolutions of 3.5 nm, 10 nm, and 7 nm at 700 nm, 1500 nm, and 2100 nm respectively. Soil reflectance was interpolated to 1 nm intervals. The instrument was calibrated with a 99% white reference panel (5 × 5 cm) every 10 samples. Samples were placed in Petri dishes and scanned with a contact reflectance probe to avoid contamination with ambient light. Each spectrum is the averaging of 30 consecutives spectra (software handled). Five replicates scans in different parts of the Petri dish were averaged to obtain a representative VIS-NIR spectrum of the sample.

### 3.5. Statistical Evaluations

The software package Statgraphics Centurion (version XVIII, Statpoint Technologies, The Plains, VA, USA) was used for statistical analysis. The following basic statistical parameters were analysed: median, mean, standard deviation (SD), extreme values (Min and Max), quartiles (lower—LQ and upper—UQ), skewness and kurtosis, and coefficient of variation (CoV). Inverse distance weighting method was used to construct maps of the spatial distribution of chemical indexes in soils by ArcGIS (10.0) software (ESRI, Redlands, CA, USA).

Mean average (MA) with a segment size of 5 was applied to all spectra before analysis. All predictions were made on MA spectra but also on spectra pre-processed with standard normal variate (SNV) and first derivative (Savitzky-Golay). Due to the relatively small number of samples (74) and to the preliminary nature of this study, full cross-validation (CV) was used all along the study.

A principal component analysis (PCA) of the spectra (MA) was used to reveal the hidden structure within the soil sample dataset. PCA allows visually exploring the spectra and having an insight into their relationships with soil properties. With regards to the number of samples and preliminary investigations partial least-squares (PLS) regression was used with full cross-validation to predict the sum of 16PAH concentrations, individual PAH concentrations, and all the investigated indexes. PLS regression is a popular technique for quantitative analysis of VIS-NIR spectra and was used to determine the best correlation between the PAHs data and the spectral data. This is a reduction dimension technique that seeks a set of latent variables by maximising the covariance between the spectra (X) and PAHs concentrations or indexes (Y). The goal of PLS regression was to predict Y from X. The predictions of the different indexes were done on the class number (e.g., for TUm four classes—1 to 4 corresponding to the following ranges of TUm values: <1; 1–10; 10–100; >100) and not on the actual values of that particular index. The predicted class numbers were then rounded to the nearest whole number. Spectra pre-processing, PCA and PLS regressions were completed with Unscrambler 10.3 (Camo Analytics, Oslo, Norway) software.

## 4. Conclusions

Rural areas are often located in the proximity of highly urbanised or industrial areas and may be exposed to high, continuous inputs of various pollutants, including PAHs, which may persist for a long-time in soils and pose a threat to ecosystems and human health. In this study, we demonstrated the combination of chemical indexes and VIS-NIR spectroscopy for first-tier screening assessment and identification of areas where site-specific assessment is needed. Additionally, for the rating of the degree of risk, different criteria were used. The 16PAH concentration ranged from 311 to 316.1 × 10^3^ µg kg^−1^ and exceeded the current regulatory limit for uncontaminated soils in almost 80% of samples. The highest PAH levels were found in the central and north part of the study region in the vicinity of PAH sources (coke plant, coal mining landfill, coal recovery plant). It was shown that risk assessment results depend on the model used for risk index calculation and on the values established as a threshold for an unacceptable risk. In our study, HQ/HI indexes were based on the permissible concentration for only 10 individual PAHs (as recommended in Polish soil guidelines), while TU, TP and TEQ on toxicity data for 16PAH compounds. HQ and TU indexes calculated for individual PAHs indicated a high risk in 62–66% of the study area, whereas risk indexes obtained for PAHs mixture (TUm, TPm and TEQ) had the highest values (above acceptable limits) only in 22–55% of soil samples. Our approach based on total levels of PAHs may lead to risk overestimation, however, as a highly protective measure was useful to delineate area for detailed higher tier evaluations.

VIS-NIR spectroscopy was also used for prediction of 16PAH concentrations and environmental risk. Our results shown that VIS-NIR is a promising and helpful technique which can be used for the screening assessment of PAH contaminated soils. Since it was difficult to predict individual soil PAH concentrations, the PLS model on specific index classes was used. The model accuracy ranged from 20 to 95.9%, depending on the class and type of index tested. Generally, risk indexes prediction gave better results when binary models were applied. The accuracies of binary models were 85.1%, 87.9%, 71.6% and 94.6% for HI, TUm, TPm and TEQ indexes, respectively. For better prediction of total PAH concentrations and to develop more accurate models of risk indexes classification, more investigations are needed by including more samples of each class. However, VIS-NIR technique can be successfully applied in the first-tier risk assessment to differentiate if a sample is contaminated/non-contaminated, or there is a risk/no risk.

## Figures and Tables

**Figure 1 molecules-25-03151-f001:**
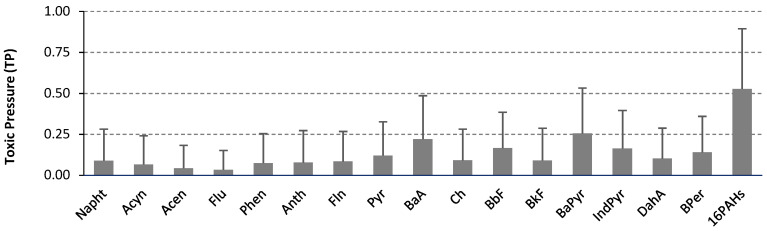
Average toxic pressure (TP) for the individual hydrocarbons and for the mixture of 16PAHs (n = 74), bar indicates the standard deviation.

**Figure 2 molecules-25-03151-f002:**
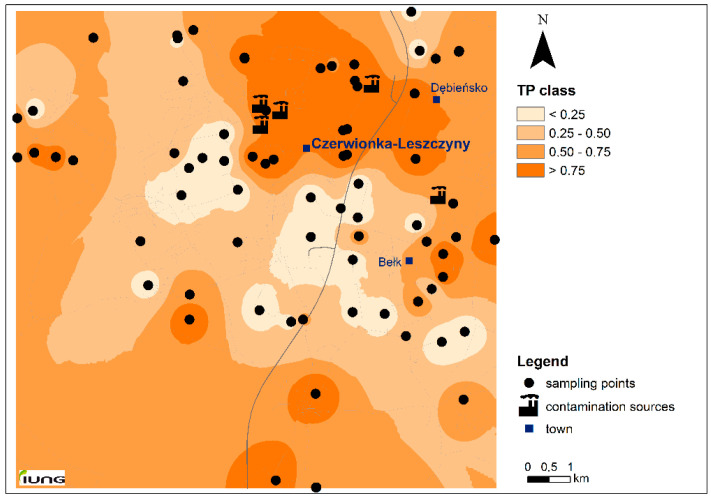
Spatial distribution of the toxic pressure (TPm) for 16PAHs in the study area; TP class limits according Dagnino et al. [15].

**Figure 3 molecules-25-03151-f003:**
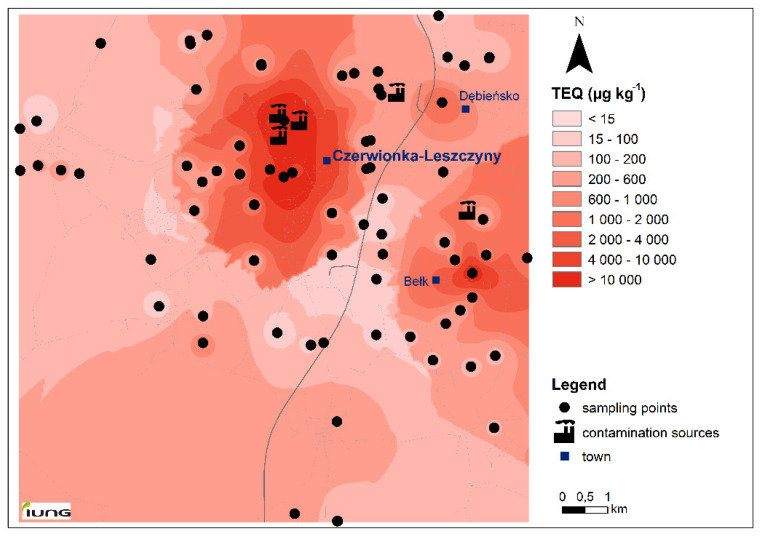
Spatial distribution of the toxic equivalent concentration (TEQ) of 16PAHs in the study area; TEQ class limits correspond to different guidelines values for the protection of human health.

**Figure 4 molecules-25-03151-f004:**
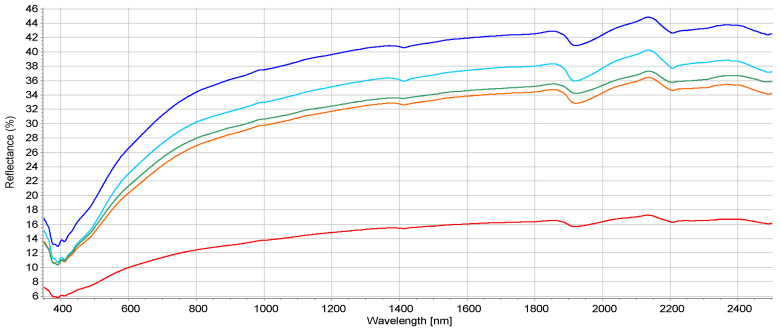
Mean reflectance spectra according to Σ16PAH content. Blue: class 1 (200–600 μg kg^−1^); turquoise: class 2 (600–1000 μg kg^−1^); green: class 3 (1000–5000 μg kg^−1^); orange: class 4 (5000–10,000 μg kg^−1^); red: class 5 (>10,000 μg kg^−1^).

**Figure 5 molecules-25-03151-f005:**
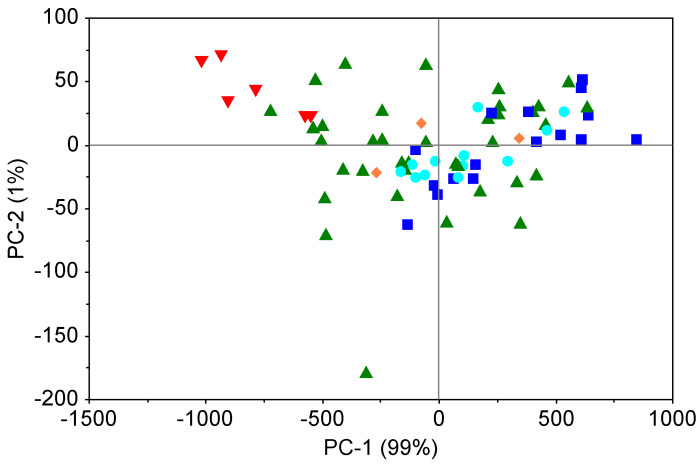
Score plot of the PCA based on soil spectral data. The colours represent PAH content. Blue: class 1 (200–600 μg kg^−1^); turquoise: class 2 (600–1000 μg kg^−1^); green: class 3 (1000–5000 μg kg^−1^); orange: class 4 (5000–10000 μg kg^−1^); red: class 5 (>10000 μg kg^−1^).

**Figure 6 molecules-25-03151-f006:**
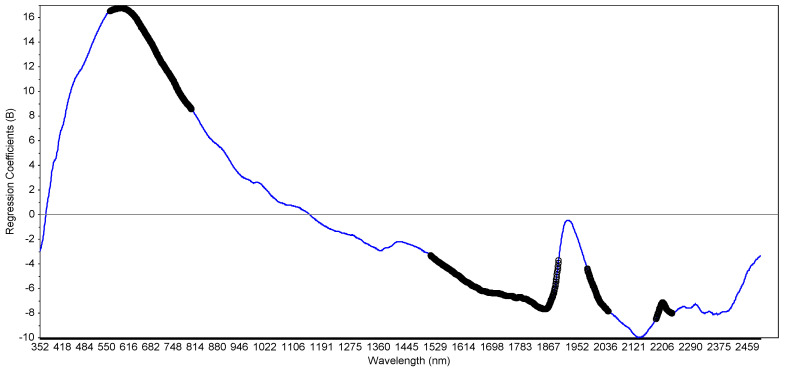
Plot of regression coefficients obtained from PLS regression of sum 16PAH. The thick black line represents the important variables regions (wavelengths) for prediction.

**Table 1 molecules-25-03151-t001:** Descriptive statistics of basic soil properties, PAH concentrations and selected PAH isomer ratios in soils (n = 74).

Variable	Min	LQ	Mean	Median	UQ	Max	CoV
**Basic soil Properties**							
Sand (%)	49.0	67.0	71.8	70.7	78.2	90.6	12
Silt (%)	9.0	21.0	26.2	26.8	31.1	45.0	30
Clay (%)	0.0	1.0	1.9	1.9	2.7	6.0	73
C_org_ (g kg^−1^)	5.2	9.5	22.9	11.7	17.6	187.2	142
pH_KCl_	3.8	4.6	5.3	5.2	5.6	7.8	17
TN (g kg^−1^)	0.2	0.8	1.5	1.0	1.4	11.7	112
**PAHs** **(** **µg kg^−1^)**							
2-ring	19	45	297	61	94	5.4 × 10^3^	325
3-ring	45	100	1631	173	313	40.2 × 10^3^	392
4-ring	99	281	5460	569	1112	137.7 × 10^3^	407
5-ring	55	144	2837	286	480	81.7 × 10^3^	430
6-ring	26	89	1566	157	247	51.5 × 10^3^	446
ΣPAH_Carcin_	137	275	5383	543	950	153.6 × 10^3^	427
ΣPAH_4em_	37	134	2976	262	435	105.2 × 10^3^	466
Σ16PAH	311	624	11792	1252	2148	316.1 × 10^3^	410
**Isomer Ratios**							
Fln/(Fln + Pyr)	0.41	0.57	0.58	0.58	0.59	0.67	5.4
BaA/(BaA + Ch)	0.26	0.37	0.41	0.43	0.46	0.53	14.5
IndPyr/(IndPyr + BPer)	0.17	0.49	0.51	0.52	0.53	0.80	12.2

Sand—fraction of 2.0–0.05 mm; silt—fraction of 0.05–0.002 mm; clay—fraction < 0.002 mm; C_org_—total organic carbon content; TN—total nitrogen content; Min—minimum value; Max—maximum value; LQ—lower quartile; UQ—upper quartile; CoV—coefficient of variation (%); ΣPAH_Carcin_—concentration of carcinogenic PAHs (BaA, Ch, BbF, BkF, BaPyr, IndPyr, DahA); ΣPAH_4em_—concentration of PAHs (BbF, BkF, BaPyr, IndPyr) derived from the emission.

**Table 2 molecules-25-03151-t002:** Descriptive statistics of hazard quotient (HQ) for individual PAH compounds and the acceptable soil screening level (MPC, µg kg^−1^) used for HQs calculation; additionally, percentage of soil samples with HQ > 1 is presented.

PAH	MPC	HQ							HQ > 1 (%)
		Min	LQ	Mean	Median	UQ	Max	CoV	
Napht	100	0.19	0.45	2.97	0.61	0.94	54.3	325	24
Anth	200	0.01	0.03	1.06	0.06	0.11	33.2	458	7
BaA	100	0.14	0.31	8.56	0.69	1.26	245.9	436	38
Ch	200	0.10	0.26	4.22	0.47	0.90	108.2	404	23
BbF	100	0.09	0.82	11.12	1.32	2.40	258.3	396	62
BkF	100	0.09	0.25	6.37	0.54	0.82	200.1	462	20
BaPyr	100	0.10	0.41	10.89	0.96	1.38	407.5	479	45
IndPyr	200	0.05	0.21	3.84	0.37	0.59	117.4	439	18
DahA	100	0.01	0.07	0.77	0.14	0.25	22.1	382	8
BPer	200	0.04	0.19	3.61	0.34	0.51	129.1	465	16
HI	-	1.56	3.06	53.41	5.42	9.21	1522.9	422	23

HI—hazard index; Min—minimum value; Max—maximum value; LQ—lower quartile; UQ—upper quartile; CoV—coefficient of variation (%); MPC—maximum permissible concentration of PAHs according to Polish soil guidelines [48].

**Table 3 molecules-25-03151-t003:** Descriptive statistics of toxic units (TU) for individual PAH compounds and the predicted no effect concentrations (PNEC, µg kg^−1^) used for TUs calculation; additionally, percentage of soil samples with TU and TUm > 1 was shown.

PAH	PNEC	TU							TU > 1 (%)
		Min	LQ	Mean	Median	UQ	Max	CoV	
Napht	1000	0.019	0.045	0.30	0.06	0.09	5.43	325	5
Acyn	290	0.005	0.009	0.18	0.02	0.03	4.85	433	4
Acen	38	0.078	0.187	2.27	0.32	0.51	62.77	402	12
Flu	1000	0.003	0.008	0.12	0.01	0.02	2.46	394	4
Phen	1800	0.019	0.042	0.64	0.07	0.13	15.19	383	7
Anth	130	0.019	0.041	1.63	0.09	0.17	51.08	458	8
Fln	1500	0.025	0.079	1.39	0.16	0.31	31.43	399	7
Pyr	1000	0.027	0.079	1.68	0.18	0.35	44.34	412	7
BaA	79	0.182	0.390	10.83	0.88	1.59	311.37	436	43
Ch	550	0.035	0.093	1.54	0.17	0.33	39.36	404	8
BbF	280	0.033	0.293	3.97	0.47	0.86	92.25	395	23
BkF	270	0.035	0.092	2.36	0.20	0.30	74.11	462	8
BaPyr	53	0.184	0.770	20.55	1.82	2.60	768.87	480	66
IndPyr	130	0.081	0.319	5.90	0.56	0.91	180.62	439	22
DahA	54	0.027	0.127	1.43	0.26	0.45	40.87	382	14
BPer	170	0.050	0.224	4.24	0.40	0.60	151.86	465	22
TUm	-	1.26	2.88	59.04	5.70	9.51	1858.24	439	100

TUm—toxic units calculated for the mixture of 16PAHs; Min—minimum value; Max—maximum value; LQ—lower quartile; UQ—upper quartile; CoV—coefficient of variation (%); PNEC—predicted no-effect concentrations [52].

**Table 4 molecules-25-03151-t004:** Percentage share of soil samples in individual classes of toxic pressure (TP); TP classification according to Dagnino et al. [15]; MPC—maximum permissible concentration in µg kg^−1^ according to Verbruggen [54].

PAH	MPC	Soil Samples in TP Class (%)			
		TP < 0.25	0.25 < TP < 0.50	0.50 < TP < 0.75	0.75 < TP < 1
Napht	690	93.2	1.4	1.4	4.1
Acyn	170	93.2	2.7	1.4	2.7
Acen	680	95.9	0.0	2.7	1.4
Flu	1600	95.9	0.0	4.1	0.0
Phen	3600	91.9	4.1	0.0	4.1
Anth	340	90.5	4.1	1.4	4.1
Fln	4800	93.2	2.7	0.0	4.1
Pyr	1800	87.8	5.4	2.7	4.1
BaA	190	66.2	17.6	8.1	8.1
Ch	1600	91.9	4.1	0.0	4.1
BbF	790	81.1	12.2	2.7	4.1
BkF	790	91.9	2.7	1.4	4.1
BaPyr	160	60.8	20.3	10.8	8.1
IndPyr	380	79.7	12.2	2.7	5.4
DahA	180	90.5	4.1	1.4	4.1
BPer	490	83.8	8.1	2.7	5.4
Σ16PAH	-	33.8	10.8	20.3	35.1

**Table 5 molecules-25-03151-t005:** Descriptive statistics of toxic equivalent concentration (TEQ, µg kg^−1^) for individual PAH compounds and the toxic equivalent factors (TEFs) used for TEQ calculation.

PAHs	TEFs	TEQ						
		Min	LQ	Mean	Median	UQ	Max	CoV
Napht	0.001	0.02	0.04	0.30	0.06	0.09	5.4	325
Acyn	0.001	0.001	0.003	0.05	0.01	0.01	1.4	433
Acen	0.001	0.003	0.007	0.09	0.01	0.02	2.4	402
Flu	0.001	0.003	0.008	0.12	0.01	0.02	2.5	394
Phen	0.001	0.03	0.08	1.16	0.13	0.23	27.3	382
Anth	0.01	0.02	0.05	2.12	0.12	0.22	66.4	457
Fln	0.001	0.04	0.12	2.08	0.24	0.47	47.1	399
Pyr	0.001	0.03	0.08	1.68	0.18	0.35	44.3	412
BaA	0.1	1.44	3.08	85.59	6.93	12.57	2459.8	436
Ch	0.01	0.19	0.51	8.45	0.94	1.81	216.5	404
BbF	0.1	0.91	8.19	111.08	13.24	23.98	2583.1	395
BkF	0.1	0.93	2.47	63.73	5.37	8.20	2000.9	462
BaPyr	1	9.77	40.81	1089.4	96.25	138.0	40,750.3	479
IndPyr	0.1	1.06	4.14	76.77	7.32	11.8	2348.1	439
DahA	1	1.48	6.84	77.11	13.96	24.50	2207.0	381
BPer	0.01	0.08	0.38	7.22	0.69	1.02	258.2	465
Σ16PAH	-	21	68	1527	151	225	52,531	456

Min—minimum value; Max—maximum value; LQ—lower quartile; UQ—upper quartile; CoV—coefficient of variation (%); TEF—values according to Nisbet and LaGoy [55].

**Table 6 molecules-25-03151-t006:** Results of indexes prediction with PLS regression.

Index	Criterion	Class Number						
		FD	1	2	3	4	5	i
Σ16PAH	A	58.1%	25.0%	75.0%	64.9%	66.7%	66.7%	*
Σ16PAH	<600 µg kg^−1^	79.7%	-	-	-	-	-	
HI	B	83.8%	92.8%	61.5%	50%	-	-	
HI	< 10	85.1%	-	-	-	-	-	
TUm	C	83.8%	91.2%	53.8%	75%	-	-	
TUm	TUm < 10	87.9%	-	-	-	-	-	
TPm	D	33.8%	20%	75%	26.7%	38.5%	-	
TPm	TPm < 0.5	71.6%	-	-	-	-	-	
TEQ	E	70.3%	-	-	-	-	-	<100
TEQ	E	79.7%	-	-	-	-	-	<200
TEQ	E	94.6%	-	-	-	-	-	<600
TEQ	E	94.6%	-	-	-	-	-	<1000
TEQ	E	94.6%	-	-	-	-	-	<2000
TEQ	E	94.6%	-	-	-	-	-	<4000
TEQ	E	95.9%	-	-	-	-	-	<10,000

HI—hazard index; TUm—toxic unit for 16PAH mixture; TPm—toxic pressure coefficient; TEQ—toxic equivalency quotient; Criterion—Classification used or particular threshold from a classification; FD—full dataset (predictions of all indexes in one run or binary prediction at a particular threshold); i—remarks (missing class or threshold for human health); * the class 0 (16PAHs < 200 µg kg^−1^) was not represented in our dataset; all threshold in µg kg^−1^; A—classification from Maliszewska-Kordybach [45] and Terelak et al. [46] for 16PAHs; B—classification from Moreno-Jimenez et al. [17]; C—from Persoone et al. [53]; D—from Dagnino et al. [15]; E—from Carlon et al. [58] and Cachada et al. [3].

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
