# Peer review of "Screening Risk Assessment of Agricultural Areas under a High Level of Anthropopressure Based on Chemical Indexes and VIS-NIR Spectroscopy"

_molecules, 2020, doi:10.3390/molecules25143151_

Round 1

Reviewer 1 Report

The MS “Screening risk assessment of agricultural area under high level of  anthropopressure based on chemical indexes and VIS-NIR spectroscopy” by Agnieszka Klimkowicz-Pawlas and Guillaume Debaene, reports the investigation of various pollutants in different locations around Czerwionka municipality, Poland, also exploring the detection limits of VIS-NIR spectroscopy. The MS is well written, results of the screening are robust and compared adequately with similar studies. The proposed usage of VIS-NIR spectroscopy for diagnostic risk screening of PAH contaminated soils, and for prediction of their toxicity based on various risk indexes is less supported from the data obtained; however, this section also represents an interesting contribution which needs to be validated by further studies.

Comments:

Line 25 “Prediction of risk indexes for the first time was based on spectral data using the PLS model, better results were obtained with binary model.” This sentence needs to be reformulated.

Line 31 “Different soil functions and services”. I am not sure I understand what kind of “soil services” are affected by pollutants. Maybe just mention soil functions.

Lines 40-42. “Major sources of PAH contamination are a result of anthropogenic activities and include biomass combustion, domestic and industrial coal combustion, coking industry and emissions from road vehicles and contribute to a continuous input of these contaminants.” Long sentence which requires revision.

Suggestion: “Major sources of PAH contamination are a result of anthropogenic activities and include biomass combustion, domestic and industrial coal combustion, coking industry and emissions from road vehicles; all these sources contribute to a continuous input of PAH contaminants.”

Line 45:” PAHs contaminating agricultural soils attract” should be “PAHs contaminated agricultural soils attract”

Line 48. “Given the pressure on soil for food security”. The pressure in not on soil but on industry, farmers and control agencies to keep the soils healthy.

Line 81: The sentence containing “… more recent studies have demonstrated for heavy metals …” needs attention. Maybe the insertion of a few words will improve its reading: ”more recent studies have demonstrated its efficacy in the detection of heavy metals…”

Line 93. Move the comma after both or remove it.        

Line 208 “corresponds to the low European values according to the European Union criteria”. Are these the lower limit values for Tn and Corg? If yes, please use “lower limit values” or “lower range according to ….”

Line 351 and 382. Figures 2 and 3. The map could be slightly improved. It was hard to discover the cardinal point indicating N. Also, the bar indicating 1 km could be labeled with larger fonts as it was hard to read 0, 0.5 and 1 after zooming the figure at 150%.

Lines 352-353 and 383-385. Legends of figures 2 and 3.

Legend of figure 2. “Spatial distribution of the toxic pressure (TPm) for 16PAHs in the study area, the results were interpolated using inverse distance weighting method; TP class limits according Dagnino et al.”. I suggest having either a full stop or semicolon after “the study area”, not a comma.

I suggest the same for the legend of Figure 3. “Spatial distribution of the toxic equivalent concentration (TEQ) of 16PAHs in the study area, the results were interpolated using inverse distance weighting method; TEQ class limits correspond to different guidelines values for protection of human health.” The sentence has to be split in at least two sentences.

Line 398. Is not the “small peak” at 2210 nm more evident than the peak at 1400 nm? Are these inverted peaks because of the plotting?

Line 412. Figure 5. Good separation on PCA plot can be seen only between extremes, i.e., class 1 and class 5 but not between classes 1 to 4. The results suggest that if soil texture is different, samples are dry, and 16PAH are not very different, the method is not able to discriminate PAH load. It is very surprising that class 4 was quite different from class 5 but could not be separated from class 1 and 2, though the latter classes had a much lower PAH content.

Line 445 and 448. “Black circles”? At 100% one can see only thick black lines. If the figure is zoomed at 150% one can guess some circles in the 1867 nm area. Maybe another way to describe/label these areas is needed.

Line 457. What kind of PAH prediction? Presence/absence?

Reviewer 2 Report

The study by Klimkowicz-Pawlas and Debaene develop an approach using vis NIR to evaluate the risk assessment and contamination prediction in soils of Poland. In particular, they found 78% of the soils contaminated and that 62% of the study area needs further action. Additionally, they tested prediction of these contaminants as a function of vis NIR spectroscopy to product a model with >70% accuracy in the prediction of contaminants levels. The results of the study is an important one because identifying contaminant source and their levels is important to guide future risk assessment and inform decision makers for future actions. The manuscript is also well written and certainly resonates with the reader of Molecules. However, there are several minor issues that need to be address prior to publication. See my details comments below:

Major Comments:

  1. It is unclear how a samples size of 74 is good to evaluate the contamination status of a study area that covers territory of 115km2. I am particularly concerned how these sample size were picked and whether these samples sites are representative of the study territory. It would be great if the authors could provide more details on the basis of picking 74 sample size and justify how these samples represent the broader distribution of contamination status in the study area. While some details are provided in section 2.1, I feel that these samples sites are purposefully selected to pick areas with direct exposure to local and transboundary pollution emission sources.
  2. It is also unclear why measurement of soil physiochemical properties are important in this study (i.e. Table 1). Since this study is more focused on quantify PAHs, there is no basis to include the physiochemical properties of soils
  3. I am also left wondering why only the PLSR model was tested to evaluate the prediction of contaminant levels in this study. Given that there are different machine learning and statistical approaches being developed recently to predict response variables as a function of vis NIR spectra, the authors need to justify why PLSR is the best model or at least how PLSR fits better in this study given the limited sample size
  4. Also, I did not see any description on how the calibration model using PLSR was developed and how the performance of this model was evaluated against independent datasets.
  5. Figure 2 and Figure 3. I think this is the weak part of the manuscript. Using these 74 sample points to interpolate over the whole study region need to be justified.
